# MULTI-DOMAIN ADVERSARIAL LEARNING

**Alice Schoenauer Sebag**[1,†]
alice.schoenauer@polytechnique.org

**Louise Heinrich**[1],
louise.heinrich@ucsf.edu

**Marc Schoenauer**[2],
marc.schoenauer@inria.fr

**Michele Sebag**[2],
sebag@lri.fr

**Lani F. Wu**[1]
lani.wu@ucsf.edu

**Steven J. Altschuler**[1]
steven.altschuler@ucsf.edu

[1] **Department of Pharmaceutical Chemistry**
UCSF, San Francisco, CA 94158

[2] **INRIA-CNRS-UPSud-UPSaclay**
TAU, U. Paris-Sud, 91405 Orsay

## ABSTRACT

Multi-domain learning (MDL) aims at obtaining a model with minimal average risk across multiple domains. Our empirical motivation is automated microscopy data, where cultured cells are imaged after being exposed to known and unknown chemical perturbations, and each dataset displays significant experimental bias. This paper presents a multi-domain adversarial learning approach, MULANN, to leverage multiple datasets with overlapping but distinct class sets, in a semi-supervised setting. Our contributions include: i) a bound on the average- and worst-domain risk in MDL, obtained using the $\mathcal{H}$-divergence; ii) a new loss to accommodate semi-supervised multi-domain learning and domain adaptation; iii) the experimental validation of the approach, improving on the state of the art on three standard image benchmarks, and a novel bioimage dataset, CELL.[1]

## 1 INTRODUCTION

Advances in technology have enabled large scale dataset generation by life sciences laboratories. These datasets contain information about overlapping but non-identical known and unknown experimental conditions. A challenge is how to best leverage information across multiple datasets on the same subject, and to make discoveries that could not have been obtained from any individual dataset alone.

Transfer learning provides a formal framework for addressing this challenge, particularly crucial in cases where data acquisition is expensive and heavily impacted by experimental settings. One such field is automated microscopy, which can capture thousands of images of cultured cells after exposure to different experimental perturbations (e.g from chemical or genetic sources). A goal is to classify mechanisms by which perturbations affect cellular processes based on the similarity of cell images. In principle, it should be possible to tackle microscopy image classification as yet another visual object recognition task. However, two major challenges arise compared to mainstream visual object recognition problems (Russakovsky et al., 2015). First, biological images are heavily impacted by experimental choices, such as microscope settings and experimental reagents. Second, there is no standardized set of labeled perturbations, and datasets often contain labeled examples for a subset of possible classes only. This has limited microscopy image classification to single datasets and does not leverage the growing number of datasets collected by the life sciences community. These challenges make it desirable to learn models across many microscopy datasets, that achieve both good robustness w.r.t. experimental settings and good class coverage, all the while being robust to the fact that datasets contain samples from overlapping but distinct class sets.

---

† Now at the French Ministry for the Economy and Finance, 75012 Paris.
[1]Code and data: github.com/AltschulerWu-Lab/MuLANN

Multi-domain learning (MDL) aims to learn a model of minimal risk from datasets drawn from distinct underlying distributions (Dredze et al., 2010), and is a particular case of transfer learning (Pan & Yang, 2010). As such, it contrasts with the so-called domain adaptation (DA) problem (Bickel et al., 2007; Ben-David et al., 2010; Ganin et al., 2016; Pan & Yang, 2010). DA aims at learning a model with minimal risk on a distribution called "target" by leveraging other distributions called "sources". Notably, most DA methods assume that target classes are identical to source classes, or a subset thereof in the case of partial DA (Cao et al., 2018; Zhang et al., 2018).

The expected benefits of MDL, compared to training a separate model on each individual dataset, are two-fold. First, MDL leverages more (labeled and unlabeled) information, allowing better generalization while accommodating the specifics of each domain (Dredze et al., 2010; Xiao et al., 2016). Thus, MDL models have a higher chance of *ab initio* performing well on a new domain − a problem referred to as domain generalization (Muandet et al., 2013) or zero-shot domain adaptation (Yang & Hospedales, 2015). Second, MDL enables knowledge transfer between domains: in unsupervised and semi-supervised settings, concepts learned on one domain are applied to another, significantly reducing the need for labeled examples from the latter (Pan & Yang, 2010).

Learning a single model from samples drawn from $n$ distributions raises the question of available learning guarantees regarding the model error on each distribution. Kifer et al. (2004) introduced the notion of $\mathcal{H}$-divergence to measure the distance between source and target marginal distributions in DA. Ben-David et al. (2006; 2010) have shown that a finite sample estimate of this divergence can be used to bound the target risk of the learned model.

The contributions of our work are threefold. First, we extend the DA guarantees to MDL (Sec. 3.1), showing that the risk of the learned model over all considered domains is upper bounded by the oracle risk and the sum of the $\mathcal{H}$-divergences between any two domains. Furthermore, an upper bound on the classifier imbalance (the difference between the individual domain risk, and the average risk over all domains) is obtained, thus bounding the worst-domain risk. Second, we propose the approach *Multi-domain Learning Adversarial Neural Network* (MuLann), which extends Domain Adversarial Neural Networks (DANNs) (Ganin et al., 2016) to semi-supervised DA and MDL. Relaxing the DA assumption, MuLann handles the so-called class asymmetry issue (when each domain may contain varying numbers of labeled and unlabeled examples of a subset of all possible classes), through designing a new loss (Sec. 3.2). Finally, MuLann is empirically validated in both DA and MDL settings (Sec. 4), as it significantly outperforms the state of the art on three standard image benchmarks (Saenko et al., 2010; Le Cun et al., 1998), and a novel bioimage benchmark, Cell, where the state of the art involves extensive domain-dependent pre-processing.

**Notation.** Let $\mathcal{X}$ denote an input space and $\mathcal{Y} = \{1, \ldots, L\}$ a set of classes. For $i = 1, \ldots, n$, dataset $S_i$ is an iid sample drawn from distribution $\mathcal{D}_i$ on $\mathcal{X} \times \mathcal{Y}$. The marginal distribution of $\mathcal{D}_i$ on $\mathcal{X}$ is denoted by $\mathcal{D}_i^{\mathcal{X}}$. Let $\mathcal{H}$ be a hypothesis space; for each $h$ in $\mathcal{H}$ ($h : \mathcal{X} \mapsto \mathcal{Y}$) we define the risk under distribution $\mathcal{D}_i$ as $\epsilon_i(h) = \mathbb{P}_{\mathbf{x}, y \sim \mathcal{D}_i}(h(\mathbf{x}) \neq y)$. $h_i^\star$ (respectively $h^\star$) denotes the oracle hypothesis according to distribution $\mathcal{D}_i$ (resp. with minimal total risk over all domains):

$$\epsilon_i^\star = \epsilon_i(h_i^\star) = \min_{h \in \mathcal{H}} \epsilon_i(h) \tag{1}$$

$$\bar{\epsilon}(h^\star) = \min_{h \in \mathcal{H}} \bar{\epsilon}(h) = \min_{h \in \mathcal{H}} \frac{1}{n} \sum_i \epsilon_i(h) \tag{2}$$

In the semi-supervised setting, the label associated with an instance might be missing. In the following, "domain" and "distribution" will be used interchangeably, and the "classes of a domain" denote the classes for which labeled or unlabeled examples are available in this domain.

## 2 STATE OF THE ART

Machine learning classically relies on the iid setting: when training and test samples are independently drawn from the same joint distribution $P(X, Y)$ (Vapnik, 1998). Two other settings emerged in the 1990s, "concept drift" and "covariate shift". They respectively occur when conditional data distributions $P(Y|X)$ and marginal data distributions $P(X)$ change, either continuously or abruptly, across training data or between train and test data (Shimodaira, 2000). Since then, transfer learning has come to designate methods to learn across drifting, shifting or distinct distributions, or even distinct tasks (Pratt et al., 1991; Pan & Yang, 2010). Restricting ourselves to addressing a single

task on a common input space, we distinguish two objectives: minimizing the learning risk over *all* considered distributions (MDL), or over *a single* target distribution while exploiting samples from richer source(s) (DA). MDL is thus distinct from multiple source DA by their respective focus on the average risk over all distributions, versus target accuracy only. Samples from the different domains can be all, partially, or not labeled (supervised, semi-supervised and unsupervised settings). Finally, different domains can involve the same classes, or some domains can involve classes not included in other domains, referred to as *class asymmetry*.

In MDL, the different domains can be taken into account by maintaining shared and domain-specific parameters (Dredze et al., 2010), or through a domain-specific use of shared parameters. The domain-dependent use of these parameters can be learned, e.g. using domain-guided dropout (Xiao et al., 2016), or based on prior knowledge about domain semantic relationships (Yang & Hospedales, 2015).

Early DA approaches leverage source examples to learn on the target domain in various ways, e.g. through reweighting source datapoints (Mansour, 2009; Huang et al., 2006; Gong et al., 2013), or defining an extended representation to learn from both source and target (Daumé III & Marcu, 2006). Other approaches proceed by aligning the source and target representations with PCA-based correlation alignment (Sun et al., 2016), or subspace alignment (Fernando et al., 2015). In the field of computer vision, a somewhat related way of mapping examples in one domain onto the other is image-to-image translation, possibly in combination with a generative adversarial network (see references in Appendix A).

Intuitively, the difficulty of DA crucially depends on the distance between source and target distribution. Accordingly, a large set of DA methods proceed by reducing this distance in the original input space $\mathcal{X}$, e.g. via importance sampling (Bickel et al., 2007) or by modifying the source representation using optimal transport (Courty et al., 2017; Damodaran et al., 2018). Another option is to map source and target samples on a latent space where they will have minimal distance. Neural networks have been intensively exploited to build such latent spaces, either through generative adversarial mechanisms (Tzeng et al., 2017; Ghifary et al., 2016), or through combining task objective with an approximation of the distance between source(s) and target. Examples of used distances include the Maximum Mean Discrepancy due to Gretton et al. (2007) (Tzeng et al., 2014; Bousmalis et al., 2016), some of its variants (Long et al., 2015; 2016), the $\mathcal{L}_2$ contrastive divergence (Motiian et al., 2017), the Frobenius norm of the output feature correlation matrices (Sun & Saenko, 2016), or the $\mathcal{H}$-divergence (Ben-David et al., 2006; 2010; Ganin et al., 2016; Pei et al., 2018; Long et al., 2017) (more in Sec. 3). Most DA methods assume that source(s) and target contain examples from the same classes; in particular, in standard benchmarks such as OFFICE (Saenko et al., 2010), all domains contain examples from the same classes. Notable exceptions are partial DA methods, where target classes are expected to be a subset of source classes e.g. (Zhang et al., 2018; Cao et al., 2018). DA and partial DA methods share two drawbacks when applied to semi-supervised MDL with non-identical domain class sets. First, neither generic nor partial DA methods try to mitigate the impact of unlabeled samples from a class without any labeled counterparts. Second, as they focus on target performance, (partial) DA methods do not discuss the impact of extra labeled source classes on source accuracy. However, as shown in Sec. 4.3, class asymmetry can heavily impact model performance if not accounted for.

Bioinformatics is increasingly appreciating the need for domain adaptation methods (Borgwardt et al., 2006; Schweikert et al., 2008; Xu & Yang, 2011; Vallania et al., 2017). Indeed, experimentalists regularly face the issues of concept drift and covariate shift. Most biological experiments that last more than a few days are subject to technical variations between groups of samples, referred to as *batch effects*. Batch effects in image-based screening data are usually tackled with specific normalization methods (Birmingham et al., 2009). More recently, work by Ando et al. (2017) applied CorAl (Sun et al., 2016) for this purpose, aligning each batch with the entire experiment. DA has been applied to image-based datasets for improving or accelerating image segmentation tasks (Becker et al., 2015; van Opbroek et al., 2015; Bermúdez-Chacón et al., 2016; Kamnitsas et al., 2017). However, to our knowledge, MDL has not yet been used in Bioimage Informatics, and this work is the first to leverage distinct microscopy screening datasets using MDL.

## 3 Multi-Domain Adversarial Learning

The $\mathcal{H}$-divergence has been introduced to bound the DA risk (Ben-David et al., 2006; 2010; Ganin et al., 2016). This section extends the DA theoretical results to the MDL case (Sec. 3.1), supporting

the design of the MuLann approach (Sec. 3.2). The reader is referred to Appendix B for formal definitions and proofs.

## 3.1 $\mathcal{H}$-DIVERGENCE FOR MDL

The distance between source and target partly governs the difficulty of DA. The $\mathcal{H}$-divergence has been introduced to define such a distance which can be empirically estimated with proven guarantees (Batu et al., 2000; Kifer et al., 2004). This divergence measures how well one can discriminate between samples from two marginals. It inspired an adversarial approach to DA (Ganin et al., 2016), through the finding of a feature space in which a binary classification loss between source and target projections is maximal, and thus their $\mathcal{H}$-divergence minimal. Furthermore, the target risk is upper-bounded by the empirical source risk, the empirical $\mathcal{H}$-divergence between source(s) and target marginals, and the oracle DA risk (Ben-David et al., 2006; 2010; Zhang et al., 2012).

**Bounding the MDL loss using the $\mathcal{H}$-divergence.** A main difference between DA and MDL is that MDL aims to minimize the average risk over all domains while DA aims to minimize the target risk only. Considering for simplicity a binary classification MDL problem and taking inspiration from (Mansour et al., 2008; Ben-David et al., 2010), the MDL loss can be formulated as an optimal convex combination of domain risks. A straightforward extension of Ben-David et al. (2010) (Theorem 2 in Appendix B.2) establishes that the compound empirical risk is upper bounded by the sum of: i) the oracle risk on each domain; ii) a statistical learning term involving the VC dimension of $\mathcal{H}$; iii) the divergence among any two domains as measured by their $\mathcal{H}$-divergence and summed oracle risk. This result states that, assuming a representation in which domains are as indistinguishable as possible and on which every 1- and 2-domain classification task is well addressed, then there exists a model that performs well on all of them. In the 2-domain case, the bound is minimized when one minimizes the convex combination of losses in the same proportion as samples.

**Bounding the worst risk.** The classifier imbalance w.r.t. the $i$-th domain is defined as $|\epsilon_i(h) - \bar{\epsilon}(h)|$. The extent to which marginal $\mathcal{D}_i$ can best be distinguished by a classifier from $\mathcal{H}$ (i.e., the $\mathcal{H}$-divergence), and the intrinsic difficulty $\epsilon_i^\star$ of the $i$-th classification task, yield an upper-bound on the classifier imbalance (proof in Appendix B.3):

**Proposition 1.** *Given an input space $\mathcal{X}$, $n$ distributions $\mathcal{D}_i$ over $\mathcal{X} \times \{0, 1\}$ and hypothesis class $\mathcal{H}$ on $\mathcal{X}$, for any $h \in \mathcal{H}$, let $\epsilon_i(h)$ (respectively $\bar{\epsilon}(h)$) denote the classification risk of $h$ w.r.t. distribution $\mathcal{D}_i$ (resp. its average risk over all $\mathcal{D}_i$). The risk imbalance $|\epsilon_i(h) - \bar{\epsilon}(h)|$ is upper bounded as:*

$$|\epsilon_i(h) - \bar{\epsilon}(h)| \leq \epsilon_i^\star + \frac{1}{n}\sum_j \epsilon_j^\star + \frac{1}{n}\sum_j \left( d_{\mathcal{H}}(\mathcal{D}_i^X, \mathcal{D}_j^X) + \Delta_{ij} \right) \tag{3}$$

*with $\Delta_{ij} = max(E_{\mathcal{D}_j^X}|h_i^\star(\mathbf{x}) - h_j^\star(\mathbf{x})|, E_{\mathcal{D}_i^X}|h_i^\star(\mathbf{x}) - h_j^\star(\mathbf{x})|)$*

Accordingly, every care taken to minimize $\mathcal{H}$-divergences or $\Delta_{ij}$ (e.g. using the class-wise contrastive losses (Motiian et al., 2017)) improves the above upper bound. An alternative bound of the classifier imbalance can be obtained by using the $\mathcal{H}\Delta\mathcal{H}$-divergence (proposition 3, and corollaries 4, 5 for the 2-domain case in Appendix).

## 3.2 MuLann: MULTI-DOMAIN ADVERSARIAL LEARNING

As pointed out by e.g. Pei et al. (2018), when minimizing the $\mathcal{H}$-divergence between two domains, a negative transfer can occur in the case of class asymmetry, when domains involve distinct sets of classes. For instance, if a domain has unlabeled samples from a class which is not present in the other domains, both global (Ganin et al., 2016) and class-wise (Pei et al., 2018) domain alignments will likely deteriorate at least one of the domain risks by putting the unlabeled samples close to labeled ones from the same domain. A similar issue arises if a domain has no (labeled or unlabeled) samples in classes which are represented in other domains. In general, unlabeled samples are only subject to constraints from the domain discriminator, as opposed to labeled samples. Thus, in the case of class asymmetry, domain alignment will tend to shuffle unlabeled samples more than labeled ones.

This limitation is addressed in MuLann by defining a new discrimination task referred to as *Known Unknown Discrimination* (KUD). Let us assume that, in each domain, a fraction $p^\star$ of unlabeled samples comes from extra classes, i.e. classes with no labeled samples within the domain. KUD aims at discriminating, within each domain, labeled samples from unlabeled ones that most likely belong to such extra classes. More precisely, unlabeled samples of each domain are ranked according to the entropy of their classification according to the current classifier, restricted to their domain classes.

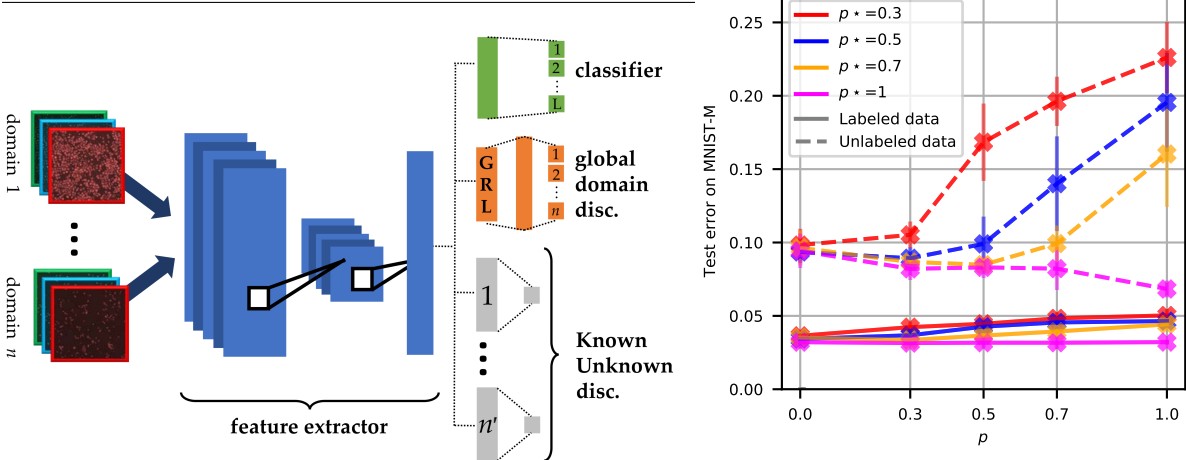

Figure 1: Left: MULANN architecture. GRL: gradient reversal layer from Ganin et al. (2016). Right: impact of parameter $p$ in comparison with the groundtruth $p^\star$ on MNIST $\to$ MNIST-M. $p = 0$ corresponds to DANN: no data flowed through the KUD module (see text for details).

Introducing the hyper-parameter $p$, the top $p\%$ examples according to this classification entropy are deemed "most likely unknown", and thus discriminated from the labeled ones of the same domain. The KUD module aims at repulsing the most likely unknown unlabeled samples from the labeled ones within each domain (Fig. 1), thus resisting the contractive effects of global domain alignment.

Overall, MULANN involves $3+n'$ interacting modules, where $n'$ is the number of domains with unlabeled data. The first module is the feature extractor with parameters $\theta_f$, which maps the input space $\mathcal{X}$ to some latent feature space $\Omega$. $2+n'$ modules are defined on $\Omega$: the classifier module, the domain discriminator module, and the $n'$ KUD modules, with respective parameters $\theta_c$, $\theta_d$ and $(\theta_{u,i})_i$. All modules are simultaneously learned by minimizing loss $\mathcal{L}(\theta_f, \theta_c, \theta_d, \theta_u)$:

$$\mathcal{L}(\theta_f, \theta_c, \theta_d, \theta_u) = \frac{1}{n}\sum_{i=1}^{n}\left(\mathcal{L}_c^i(\theta_f, \theta_c) - \lambda\,\mathcal{L}_d^i(\theta_f, \theta_d)\right) + \frac{\zeta}{n'}\sum_{j=1}^{n'}\mathcal{L}_u^j(\theta_f, \theta_{u,j}) \qquad (4)$$

where $\zeta$ and $\lambda$ are hyper-parameters, $\mathcal{L}_c^i(\theta_f, \theta_c)$ is the empirical classification loss on labeled examples in $S_i$, $\mathcal{L}_d^i(\theta_f, \theta_d)$ is the domain discrimination loss (multi-class cross-entropy loss of classifying examples from $S_i$ in class $i$), and $\mathcal{L}_u^i(\theta_f, \theta_{u,i})$ is the KUD loss (binary cross-entropy loss of discriminating labelled samples from $S_i$ from the "most likely unknown" unlabelled samples from $S_i$).

The loss minimization aims to find a saddle point $(\hat{\theta}_f, \hat{\theta}_y, \hat{\theta}_d, \hat{\theta}_u)$, achieving an equilibrium between the classification performance, the discrimination among domains (to be prevented) and the discrimination among labeled and some unlabeled samples within each domain (to be optimized). The sensitivity w.r.t. hyperparameter $p$ will be discussed in Sec. 4.3.

## 4 EXPERIMENTAL VALIDATION

This section reports on the experimental validation of MULANN in DA and MDL settings on three image datasets (Sec. 4.2), prior to analyzing MULANN and investigating the impact of class asymmetry on model performances (Sec. 4.3).

### 4.1 IMPLEMENTATION

**Datasets** The DA setting considers three benchmarks: DIGITS, including the well-known MNIST and MNIST-M (Le Cun et al., 1998; Ganin et al., 2016); Synthetic road signs and German traffic sign benchmark (Chigorin et al., 2012; Stallkamp et al., 2012) and OFFICE (Saenko et al., 2010). The MDL setting considers the new CELL benchmark, which is made of fluorescence microscopy images of cells (detailed in Appendix C). Each image contains tens to hundreds of cells that have been exposed to a given chemical compound, in three domains: California (C), Texas (T) and England (E). There are 13 classes across the three domains (Appendix, Fig. 2); a drug class is a group of compounds targeting a similar known biological process, e.g. DNA replication. Four domain shifts are considered: C↔T, T↔E, E↔C and C↔T↔E.

**Baselines and hyperparameters.** In all experiments, MuLann is compared to Dann (Ganin et al., 2016) and its extension Mada (Pei et al., 2018) (that involves one domain discriminator module per class rather than a single global one). For Dann, Mada and MuLann, the same pre-trained VGG-16 architecture (Simonyan & Zisserman, 2014) from Caffe (Jia et al., 2014) is used for Office and Cell[2]; the same small convolutional network as Ganin et al. (2016) is used for Digits (see Appendix D.1 for details). The models are trained in Torch (Collobert et al., 2011) using stochastic gradient descent with momentum ($\rho = 0.9$). As in (Ganin et al., 2016), no hyper-parameter grid-search is performed for Office results - double cross-validation is used for all other benchmarks. Hyper-parameter ranges can be found in Appendix D.2.

**Semi-supervised setting.** For Office and Cell, we follow the experimental settings from Saenko et al. (2010). A fixed number of labeled images per class is used for one of the domains in all cases (20 for Amazon, 8 for DSLR and Webcam, 10 in Cell). For the other domain, 10 labeled images per class are used for half of the classes (15 for Office, 4 for Cell). For Digits and RoadSigns, all labeled source train data is used, whereas labeled target data is used for half of the classes only (5 for Digits, 22 for RoadSigns). In DA, the evaluation is performed on all target images from the unlabeled classes. In MDL, the evaluation is performed on all source and target classes (considering labeled and unlabeled samples).

**Evaluation goals.** A first goal is to assess MuLann performance comparatively to the baselines. A second goal is to assess how the experimental setting impacts model performance. As domain discriminator and KUD modules can use both labeled and unlabeled images, a major question regards the impact of seeing unlabeled images during training. Two experiments are conducted to assess this impact: a) the same unlabeled images are used for training and evaluation (referred to as **fully transductive** setting, noted FT) ; b) some unlabeled images are used for training, and others for evaluation (referred to as **non-fully transductive** setting, noted NFT). (The case where no unlabeled images are used during training is discarded due to poor results).

### 4.2 Evaluation

**DA on** Digits**, RoadSigns and** Office**.** Table 1 compares MuLann with Dann and Mada (Sec. 4.1). Other baselines include: Learning from source and target examples with no transfer loss; Published results from (Motiian et al., 2017) (legend CCSA), that uses a contrastive loss to penalizes large (resp. small) distances between same (resp. different) classes and different domains in the feature space; Published results from (Tzeng et al., 2015), an extension of Dann that adds a loss on target softmax values ("soft label loss"; legend Tseng15). Overall, MuLann yields the best results, significantly improving upon the former best results on the most difficult cases, i.e., D→A, A→D or W→A. As could be expected, the fully transductive results match or significantly outperform the non-fully transductive ones. Notably, Mada performs similarly to Dann on Digits and RoadSigns, but worse on Office; a potential explanation is that Mada is hindered as the number of classes, and thus domain discriminators, increases (respectively 10, 32 and 43 classes).

**MDL on** Cell**.** A state of the art method for fluorescence microscopy images relies on tailored approaches for quantifying changes to cell morphology (Kang et al., 2016). Objects (cells) are segmented in each image, and circa 650 shape, intensity and texture features are extracted for each object in each image. The *profile* of each image is defined as the vector of its Kolmogorov-Smirnov statistics, computed for each feature by comparing its distribution to that of the same feature from pooled negative controls of the same plate[3]. Classification in profile space is realized using linear discriminant analysis, followed by k-nearest neighbor (LDA+k-NN) ("Baseline P" in Table 2). As a state of the art shallow approach to MDL to be applied in profile space, CORAL (Sun et al., 2016) was chosen ("P + CORAL" in Table 2). A third baseline corresponds to fine-tuning VGG-16 without any transfer loss ("Baseline NN").

Table 2 compares Dann, Mada and MuLann to the baselines, where columns 4-7 (resp. 8-9) consider raw images (resp. the profile representations).[4] The fact that a profile-based baseline generally outperforms an image-based baseline was expected, as profiles are designed to reduce the impact of experimental settings (column 4 vs. 8). The fact that standard deviations tend to be larger

---

[2]Complementary experiments with AlexNet (Krizhevsky et al., 2012) yield worse results, as already noted by (Koniusz et al., 2016).

[3]A plate contains between 96 and 384 experiments, realized the same day in exactly the same conditions.

[4]We could not obtain results with CCSA (Motiian et al., 2017) on unlabeled classes.

Table 1: Classification results on target test set in the semi-supervised DA setting (average and stdev on 5 seeds or folds). Bold: results less than 1 stdev from the best in each column. See text.

| Source
Target | | Mnist
Mnist-M | SynSigns
GTSRB | DSLR
Amazon | Amazon
DSLR | Webcam
DSLR | DSLR
Webcam | Webcam
Amazon | Amazon
Webcam | OFFICE
average |
|---|---|---|---|---|---|---|---|---|---|---|
| Baseline | | 35.6 (0.6) | 85.1 (1.2) | 35.5 (0.5) | 58.5 (1.7) | 90.9 (1.8) | 90.6 (0.6) | 34.4 (2.7) | 55.8 (1.5) | 61.0 |
| Tzeng15 | | - | - | 43.1 (0.2) | 68.0 (0.5) | 97.5 (0.1) | 90.0 (0.2) | 40.5 (0.2) | 59.3 (0.6) | 66.4 |
| CCSA | | - | - | 42.6 (0.6) | 70.5 (0.6) | 96.2 (0.3) | 90.0 (0.2) | 43.6 (1.0) | 63.3 (0.9) | 67.8 |
| | DANN | 90.4 (1.1) | 89.8 (1.1) | 50.9 (2.4) | 68.6 (4.9) | 88.8 (3.2) | **91.9** (0.7) | 48.8 (3.8) | 73.0 (2.6) | 70.3 |
| NFT | MADA | 89.9 (0.8) | 88.7 (1.0) | 44.8 (3.3) | 64.0 (3.9) | 88.2 (4.2) | 89.1 (3.4) | 44.7 (4.8) | 72.2 (3.1) | 67.2 |
| | MULANN | **91.5 (0.4)** | **92.1 (1.4)** | **57.6** (3.9) | **75.8** (3.7) | **93.3** (2.5) | 89.9 (1.6) | **54.9** (3.9) | **76.8** (3.1) | **74.7** |
| | DANN | 90.6 (1.2) | 86.7 (0.8) | 52.2 (2.2) | 77.4 (2.2) | **94.6** (1.2) | **90.7** (1.7) | 53.0 (1.9) | 74.3 (2.7) | 73.7 |
| FT | MADA | 91.0 (1.1) | 84.8 (1.6) | 51.6 (2.5) | 78.8 (3.6) | 91.7 (1.7) | 88.8 (2.3) | 53.8 (2.6) | 73.5 (2.2) | 73.0 |
| | MULANN | **92.7 (0.6)** | **89.1 (1.5)** | **63.9** (2.4) | **81.7** (1.7) | **95.4** (2.4) | **89.3** (2.8) | **64.2** (2.5) | **80.8** (2.7) | **79.2** |

Table 2: CELL test classification accuracy results on all domains (average and stdev on 5 folds), in the fully transductive setting (see table 5 in Appendix for non-transductive ones, and sections C.4, C.5 for details about image and class selection).

| Shift | Image set | # classes | Baseline NN | DANN | MADA | MULANN | Baseline P | P+Coral |
|---|---|---|---|---|---|---|---|---|
| E-C | E | 7 | 63.7 (7.0) | 62.9 (7.6) | 59.5 (9.5) | 64.4 (8.0) | 74.1 (3.9) | 58.4 (6.1) |
| | C lab. | 4 | 97.0 (1.6) | 86.4 (10.3) | 86.1 (6.5) | 82.4 (10.2) | 95.4 (3.2) | 86.6 (6.0) |
| | C unlab. | 3 | 0.6 (1.2) | 54.4 (18.3) | 33.6 (17.5) | 58.4 (19.7) | 25.5 (5.7) | 42.2 (9.5) |
| C-T | C | 10 | 90.4 (1.8) | 90.0 (1.3) | 87.2 (2.4) | 88.0 (3.6) | 96.1 (1.0) | 93.8 (0.9) |
| | T lab. | 7 | 93.8 (2.0) | 93.6 (1.8) | 89.2 (2.4) | 90.0 (1.9) | 95.2 (3.1) | 93.4 (3.0) |
| | T unlab. | 3 | 36.4 (10.7) | 68.3 (6.4) | 63.7 (10.4) | 91.6 (5.7) | 68.1 (2.1) | 86.0 (7.8) |
| T-E | T | 7 | 88.9 (6.6) | 90.8 (3.9) | 87.7 (2.1) | 85.7 (6.6) | 89.3 (8.7) | 90.3 (3.1) |
| | E lab. | 4 | 60.0 (5.3) | 59.4 (6.8) | 56.5 (12.3) | 54.5 (6.5) | 59.4 (8.1) | 50.3 (6.4) |
| | E unlab. | 3 | 19.0 (14.4) | 72.7 (10.1) | 56.2 (16.6) | 71.7 (21.9) | 32.9 (12.3) | 48.1 (10.0) |
| C-T-E | C | 7 | 89.8 (3.5) | 87.8 (4.6) | 92.8 (1.5) | 88.8 (5.2) | 96.3 (1.1) | 89.3 (5.0) |
| | T | 7 | 92.6 (2.6) | 90.2 (1.2) | 94.2 (2.3) | 92.5 (3.0) | 96.8 (2.5) | 89.9 (3.1) |
| | E lab. | 4 | 62.3 (5.5) | 56.7 (4.2) | 53.6 (8.5) | 48.1 (5.3) | 57.3 (6.1) | 44.4 (7.2) |
| | E unlab. | 3 | 19.9 (13.5) | 49.4 (6.5) | 46.5 (6.9) | 79.4 (5.3) | 45.5 (13.6) | 62.8 (7.2) |

here than for OFFICE, RoadSigns or DIGITS is explained by a higher intra-class heterogeneity; some classes comprise images from different compounds with similar but not identical biological activity. Most interestingly, MULANN and P+CORAL both improve classification accuracy on unlabeled classes at the cost of a slighty worse classification accuracy for the labeled classes (in all cases but one). This is explained as reducing the divergence between domain marginals on the latent feature space prevents the classifier from exploiting dataset-dependent biases. Overall, MULANN and P+CORAL attain comparable results on two-domain cases, with MULANN performing significantly better in the three-domain case. Finally, MULANN matches or significantly outperforms DANN and MADA.

## 4.3 ANALYSES

Two complementary studies are conducted to investigate the impact of hyperparameter $p$ and that of class asymmetry. The tSNE (van der Maaten & Hinton, 2008) visualizations of the feature space for DANN, MADA and MULANN are displayed in Appendix, Fig. 3.

**Sensitivity w.r.t. the fraction $p$ of "known unknowns".** MULANN was designed to counter the negative transfer that is potentially caused by class asymmetry. This is achieved through the repulsion of labeled examples in each domain from the fraction $p$ of unlabeled examples deemed to belong to extra classes (not represented in the domain). The sensitivity of MULANN performance to the value of $p$ and its difference to the ground truth $p^\star$ is investigated on MNIST↔MNIST-M. A first remark is that discrepancies between $p$ and $p^\star$ has no influence on the accuracy on a domain without unlabeled

Figure 3: Impact of asymmetry in class content between domains on OFFICE (W→A) for DANN, MADA and MuLANN. See text for details. Better seen in color.

| Case | Dom. 1 Lab. | Dom. 2 Lab. | Unlab. |
|---|---|---|---|
| 1 | $\alpha, \beta$ | $\alpha$ | $\beta$ |
| 2 | $\alpha, \beta, \gamma$ | $\alpha$ | $\beta$ |
| 3 | $\alpha, \beta$ | $\alpha$ | $\beta, \delta$ |
| 4 | $\alpha, \beta, \gamma$ | $\alpha$ | $\beta, \delta$ |

Table 3: Class content per case in the asymmetry experiments

datapoints (Fig. 4 in Appendix). Fig. 1, right, displays the error depending on $p$ for various values of $p^\star$. As could have been expected, it is better to underestimate than to overestimate $p^\star$; it is even better to slightly underestimate it than to get it right, as the entropy ranking of unlabeled examples can be perturbed by classifier errors.

**Impact of class/domain asymmetry.** Section 4.2 reports on the classification accuracy when all classes are represented in all domains of a given shift. In the general case however, the classes represented by the unlabeled examples are unknown, hence there might exist "orphan" classes, with labeled or unlabeled samples, unique to a single domain. The impact of such orphan classes, referred to as class asymmetry, is investigated in the 2-domain case. Four types of samples are considered (Table 3): A class might have labeled examples in both domains ($\alpha$), labeled in one domain and unlabeled in the other domain ($\beta$), labeled in one domain and absent in the other one (orphan $\gamma$), and finally unlabeled in one domain and absent in the other one (orphan $\delta$). The impact of the class asymmetry is displayed on Fig. 3, reporting the average classification accuracy of $\alpha, \beta$ classes on domain 1 on the x-axis, and classification accuracy of unlabeled $\beta$ classes on domain 2 on the y-axis, for MuLANN, DANN and MADA on OFFICE (on CELL in Fig. 5, Appendix).

A clear trend is that adding labeled orphans $\gamma$ (case "2", Fig. 3) entails a loss of accuracy for all algorithms compared to the no-orphan reference (case "1"). This is explained as follows: on the one hand, the $\gamma$ samples are subject to the classifier pressure as all labeled samples; on the other hand, they must be shuffled with samples from domain 2 due to the domain discriminator(s) pressure. Thus, the easiest solution is to shuffle the unlabeled $\beta$ samples around, and the loss of accuracy on these $\beta$ samples is very significant (the "2" is lower on the $y$-axis compared to "1" for all algorithms). The perturbation is less severe for the labeled $(\alpha, \beta)$ samples in domain 1, which are preserved by the classifier pressure ($x$-axis).

The results in case "3" are consistent with the above explanation: since the unlabeled $\delta$ samples are only seen by the discriminator(s), their addition has little impact on either the labeled or unlabeled data classification accuracy (Figs. 3 and 5). Finally, there is no clear trend in the impact of both labeled and unlabeled orphans (case "4"): labeled $(\alpha, \beta)$ (resp. unlabeled $\beta$) are only affected for MADA on CELL (resp. MuLANN on OFFICE). Overall, these results show that class asymmetry matters for practical applications of transfer learning, and can adversely affect all three adversarial methods (Figs. 3 and 5), with asymmetry in labeled class content ("2") being the most detrimental to model performance.

## 5 DISCUSSION AND FURTHER WORK

This paper extends the use of domain adversarial learning to multi-domain learning, establishing how the $\mathcal{H}$-divergence can be used to bound both the risk across all domains and the worst-domain risk (imbalance on a specific domain). The stress is put on the notion of class asymmetry, that is, when some domains contain labeled or unlabeled examples of classes not present in other domains. Showing the significant impact of class asymmetry on the state of the art, this paper also introduces MuLANN, where a new loss is meant to resist the contractive effects of the adversarial domain discriminator and to repulse (a fraction of) unlabeled examples from labeled ones in each domain.

The merits of the approach are satisfactorily demonstrated by comparison to DANN and MADA on DIGITS, RoadSigns and OFFICE, and results obtained on the real-world CELL problem establish a new baseline for the microscopy image community.

A perspective for further study is to bridge the gap between the proposed loss and importance sampling techniques, iteratively exploiting the latent representation to identify orphan samples and adapt the loss while learning. Further work will also focus on how to identify and preserve relevant domain-specific behaviours while learning in a domain adversarial setting (e.g., if different cell types have distinct responses to the same class of perturbations).

ACKNOWLEDGMENTS

This work was supported by NIH RO1 CA184984 (LFW), R01GM112690 (SJA) and the Institute of Computational Health Sciences at UCSF (SJA and LFW). We thank the Shoichet lab (UCSF) for access to their GPUs and Theresa Gebert for suggestions and feedback.

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

## A  EXTENDED STATE-OF-THE-ART: IMAGE TRANSLATION

In the field of computer vision, another way of mapping examples in one domain onto the other domain is image-to-image translation. In the supervised case (the true pairs made of an image and its translation are given), Pic2Pix (Isola et al., 2017) trains a conditional GAN to discriminate true pairs from fake ones. In the unsupervised case, another loss is designed to enforce cycle consistency (simultaneously learning the mapping $\phi$ from domain $A$ to $B$, $\psi$ from $B$ to $A$, and requiring $\phi o \psi =$Id) (Zhu et al., 2017; Yi et al., 2017). Note that translation approaches do not *per se* address domain adaptation as they are agnostic w.r.t. the classes. Additional losses are used to overcome this limitation: Domain transfer network (DTN) (Taigman et al., 2016) uses an auto-encoder-like loss in the latent space; GenToAdapt (Sankaranarayanan et al., 2017) uses a classifier loss in the latent space; UNIT (Liu et al., 2017) uses a VAE loss.

StarGAN (Choi et al., 2018) combines image-to-image translation with a GAN, where the discriminator is trained to discriminate true from fake pairs on the one hand, and the domain on the other hand. ComboGAN (Anoosheh et al., 2017) learns two networks per domain, an encoder and a decoder. DIRT-T (Shu et al., 2018) uses a conditional GAN and a classifier in the latent space, with two additional losses, respectively enforcing the cluster assumption (the classifier boundary should not cross high density region) and a virtual adversarial training (the hypothesis should be invariant under slight perturbations of the input).
Interestingly, DA and MDL (like deep learning in general) tend to combine quite some losses; two benefits are expected from using a mixture of losses, a smoother optimization landscape and a good stability of the representation (Bousquet & Elisseeff, 2002).

## B  PROOFS

### B.1  DEFINITION OF THE $\mathcal{H}$-DIVERGENCE

**Definition.** *(Kifer et al., 2004; Ben-David et al., 2006; 2010) Given a domain $\mathcal{X}$, two distributions $\mathcal{D}$ and $\mathcal{D}'$ over that domain and a binary hypothesis class $\mathcal{H}$ on $\mathcal{X}$, the $\mathcal{H}$-divergence between $\mathcal{D}$ and $\mathcal{D}'$ is defined as:*

$$d_{\mathcal{H}}(\mathcal{D}, \mathcal{D}') = 2 \sup_{h \in \mathcal{H}} |\mathbb{P}_{\mathcal{D}}(h(\mathbf{x}) = 1) - \mathbb{P}_{\mathcal{D}'}(h(\mathbf{x}) = 1)|$$

### B.2  BOUNDING MDL LOSS USING THE $\mathcal{H}$-DIVERGENCE

**Theorem 2.** *Given an input space $\mathcal{X}$, we consider $n$ distributions $\mathcal{D}_i$ over $\mathcal{X} \times \{0; 1\}$ and a hypothesis class $\mathcal{H}$ on $\mathcal{X}$ of VC dimension $d$. Let $\alpha$ and $\gamma$ be in the simplex of dimension $n$. If $S$ is a sample of size $m$ which contains $\gamma_i m$ samples from $\mathcal{D}_i$, and $\hat{h}$ is the empirical minimizer of $\sum_i \alpha_i \hat{\epsilon}_i$ on $(S_i)_i$, then for any $\delta > 0$, with probability at least $1 - \delta$, the compound empirical error is upper bounded as:*

$$\sum_i \epsilon_i(\hat{h}) \leq \sum_i \epsilon_i^{\star} + 4nB(\alpha) + 2 \sum_{i \leq j} (\alpha_i + \alpha_j) \left( d_{\mathcal{H}}(\mathcal{D}_i^X, \mathcal{D}_j^X) + \beta_{i,j} \right) \tag{5}$$

*with*

$$B(\alpha) = \sqrt{\sum_j \frac{\alpha_j^2}{\gamma_j}} \sqrt{\frac{2d \log(2(m+1)) + \log(\frac{4}{\delta})}{m}}$$

*and*

$$\beta_{i,j} = \min_{h \in \mathcal{H}} \left( \epsilon_i(h) + \epsilon_j(h) \right)$$

A tighter bound can be obtained by replacing $d_{\mathcal{H}}(\mathcal{D}_i, \mathcal{D}_j)$ with $\frac{1}{2} d_{\mathcal{H}\Delta\mathcal{H}}(\mathcal{D}_i, \mathcal{D}_j)$. The $\mathcal{H}\Delta\mathcal{H}$-divergence (Ben-David et al., 2010) operates on the symmetric difference hypothesis space $\mathcal{H}\Delta\mathcal{H}$. However, divergence $\mathcal{H}\Delta\mathcal{H}$ does not lend itself to empirical estimation: even Ben-David et al. (2010) fall back on $\mathcal{H}$-divergence in their empirical validation.

**Proof of theorem 2** For $i, j$ we note $\beta_{i,j} = \epsilon_i(h_{i,j}^\star) + \epsilon_j(h_{i,j}^\star) = \min_{h \in \mathcal{H}} (\epsilon_i(h) + \epsilon_j(h))$. For $\alpha$ in the $n$-dimensional simplex and $h \in \mathcal{H}$, we note $\epsilon_\alpha(h) = \sum_i \alpha_i \epsilon_i(h)$.

We have for $\alpha$ in the simplex of dimension $n$, $h \in \mathcal{H}$ and $j \in \{1, \ldots, m\}$, using the triangle inequality (similarly to the proof of Theorem 4 in (Ben-David et al., 2010))

$$
\begin{aligned}
|\epsilon_\alpha(h) - \epsilon_j(h)| &= \left| \sum_i \alpha_i \left( \mathbb{E}_{\mathbf{x},y\sim\mathcal{D}_i} |h(\mathbf{x}) - y| - \mathbb{E}_{\mathbf{x},y\sim\mathcal{D}_j} |h(\mathbf{x}) - y| \right) \right| \\
&\leq \sum_i \alpha_i \left| \mathbb{E}_{\mathbf{x},y\sim\mathcal{D}_i} |h(\mathbf{x}) - y| - \mathbb{E}_{\mathbf{x},y\sim\mathcal{D}_j} |h(\mathbf{x}) - y| \right| \\
&\leq \sum_i \alpha_i \left| \mathbb{E}_{\mathbf{x},y\sim\mathcal{D}_i} |h(\mathbf{x}) - y| - \mathbb{E}_{\mathbf{x}\sim\mathcal{D}_i} |h(\mathbf{x}) - h_{i,j}^\star(\mathbf{x})| \right| \\
&\quad + \alpha_i \left| \mathbb{E}_{\mathbf{x}\sim\mathcal{D}_i} |h(\mathbf{x}) - h_{i,j}^\star(\mathbf{x})| - \mathbb{E}_{\mathbf{x}\sim\mathcal{D}_j} |h(\mathbf{x}) - h_{i,j}^\star(\mathbf{x})| \right| \\
&\quad + \alpha_i \left| \mathbb{E}_{\mathbf{x}\sim\mathcal{D}_j} |h(\mathbf{x}) - h_{i,j}^\star(\mathbf{x})| - \mathbb{E}_{\mathbf{x},y\sim\mathcal{D}_j} |h(\mathbf{x}) - y| \right| \\
&\leq \sum_i \alpha_i \left( \beta_{i,j} + d_\mathcal{H}(\mathcal{D}_i, \mathcal{D}_j) \right)
\end{aligned}
$$

The last line follows from the definitions of $\beta_{i,j}$ and $\mathcal{H}$-divergence. Thus using lemma 6 in (Ben-David et al., 2010)

$$
\begin{aligned}
\epsilon_j(\hat{h}) &\leq \epsilon_\alpha(\hat{h}) + \sum_i \alpha_i \left( \beta_{i,j} + d_\mathcal{H}(\mathcal{D}_i, \mathcal{D}_j) \right) \\
&\leq \hat{\epsilon}_\alpha(\hat{h}) + 2B(\alpha) + \sum_i \alpha_i \left( \beta_{i,j} + d_\mathcal{H}(\mathcal{D}_i, \mathcal{D}_j) \right) \\
&\leq \hat{\epsilon}_\alpha(h_j^\star) + 2B(\alpha) + \sum_i \alpha_i \left( \beta_{i,j} + d_\mathcal{H}(\mathcal{D}_i, \mathcal{D}_j) \right) \\
&\leq \epsilon_\alpha(h_j^\star) + 4B(\alpha) + \sum_i \alpha_i \left( \beta_{i,j} + d_\mathcal{H}(\mathcal{D}_i, \mathcal{D}_j) \right) \\
&\leq \epsilon_i^\star + 4B(\alpha) + 2 \sum_i \alpha_i \left( \beta_{i,j} + d_\mathcal{H}(\mathcal{D}_i, \mathcal{D}_j) \right)
\end{aligned}
$$

with

$$
B(\alpha) = \sqrt{\sum_j \frac{\alpha_j^2}{\beta_j}} \sqrt{\frac{2d \log(2(m+1)) + \log(\frac{4}{\delta})}{m}}
$$

Hence the result. $\square$

### B.3 BOUNDING DOMAIN IMBALANCE

**Proof of proposition 1** We have for $h \in \mathcal{H}$ and $j \in [1, \ldots, m]$, using the triangle inequality and the definition of $\epsilon_i^\star$ (similarly to the proof of Theorem 1 in (Ben-David et al., 2006))

$$
\begin{aligned}
\epsilon_j(h) &= \mathbb{P}_{\mathbf{x},y\sim\mathcal{D}_j} (h(\mathbf{x}) \neq y) \\
&= \mathbb{E}_{\mathbf{x},y\sim\mathcal{D}_j} |h(\mathbf{x}) - y| \\
&\leq \mathbb{E}_{\mathcal{D}_j^X} |h(\mathbf{x}) - h_j^\star(\mathbf{x})| + \mathbb{E}_{\mathcal{D}_j} |h_j^\star(\mathbf{x}) - y| \\
&\leq \mathbb{E}_{\mathcal{D}_j^X} |h(\mathbf{x}) - \frac{1}{n} \sum_i h_i^\star(\mathbf{x})| + \mathbb{E}_{\mathcal{D}_j^X} |\frac{1}{n} \sum_i h_i^\star(\mathbf{x}) - h_j^\star(\mathbf{x})| + \epsilon_j^\star \\
&\leq \frac{1}{n} \sum_i \mathbb{E}_{\mathcal{D}_j^X} |h(\mathbf{x}) - h_i^\star(\mathbf{x})| + \frac{1}{n} \sum_i \mathbb{E}_{\mathcal{D}_j^X} |h_i^\star(\mathbf{x}) - h_j^\star(\mathbf{x})| + \epsilon_j^\star
\end{aligned}
$$

We have for $i$

$$\mathbb{E}_{\mathcal{D}_j^X}|h(\mathbf{x}) - h_i^\star(\mathbf{x})| \leq \mathbb{E}_{\mathcal{D}_i^X}|h(\mathbf{x}) - h_i^\star(\mathbf{x})| + |\mathbb{E}_{\mathcal{D}_i^X}|h(\mathbf{x}) - h_i^\star(\mathbf{x})| - \mathbb{E}_{\mathcal{D}_j^X}|h(\mathbf{x}) - h_i^\star(\mathbf{x})||$$

$$\leq \epsilon_i(h) + \epsilon_i^\star + d_{\mathcal{H}}(\mathcal{D}_i^X, \mathcal{D}_j^X)$$

The second line follows from the triangle inequality and the definition of the $\mathcal{H}$-divergence. Thus

$$\epsilon_j(h) \leq \frac{1}{n}\sum_i \left(\epsilon_i(h) + \epsilon_i^\star + d_{\mathcal{H}}(\mathcal{D}_i^X, \mathcal{D}_j^X) + \mathbb{E}_{\mathcal{D}_j^X}|h_i^\star(\mathbf{x}) - h_j^\star(\mathbf{x})|\right) + \epsilon_j^\star \tag{6}$$

By symmetry we obtain

$$\frac{1}{n}\sum_i \epsilon_i(h) \leq \epsilon_j(h) + \frac{1}{n}\sum_i \left(\epsilon_i^\star + d_{\mathcal{H}}(\mathcal{D}_i^X, \mathcal{D}_j^X) + \mathbb{E}_{\mathcal{D}_i^X}|h_i^\star(\mathbf{x}) - h_j^\star(\mathbf{x})|\right) + \epsilon_j^\star$$

Thus the result. $\square$

**Proposition 3.** *Given a domain $\mathcal{X}$, $m$ distributions $\mathcal{D}_i$ over $\mathcal{X} \times \{0; 1\}$ and a hypothesis class $\mathcal{H}$ on $\mathcal{X}$, we have for $h \in \mathcal{H}$ and $j \in [1, \ldots, m]$*

$$|\epsilon_j(h) - \frac{1}{n}\sum_i \epsilon_i(h)| \leq 2\left(\epsilon_j^\star + \frac{1}{n}\sum_i \epsilon_i^\star\right) + \epsilon_j(h^\star) + \beta + \frac{1}{n}\sum_i d_{\mathcal{H}}(\mathcal{D}_i^X, \mathcal{D}_j^X) + \frac{1}{2}d_{\mathcal{H}\Delta\mathcal{H}}(\mathcal{D}_i, \mathcal{D}_j)$$

$$\tag{7}$$

*where*

$$\beta = \sum_j \epsilon_j(h^\star) = \min_{h \in \mathcal{H}}\sum_j \epsilon_j(h)$$

**Proof** For $i, j \in [i, \ldots, m]$ we have

$$\mathbb{E}_{\mathcal{D}_i}|h_i^\star(\mathbf{x}) - h_j^\star(\mathbf{x})| \leq \mathbb{E}_{\mathcal{D}_i}|h_j^\star(\mathbf{x}) - h^\star(\mathbf{x})| + \mathbb{E}_{\mathcal{D}_i}|h^\star(\mathbf{x}) - h_i^\star(\mathbf{x})|$$

$$\leq \mathbb{E}_{\mathcal{D}_j}|h_j^\star(\mathbf{x}) - h^\star(\mathbf{x})| + \frac{1}{2}d_{\mathcal{H}\Delta\mathcal{H}}(\mathcal{D}_i, \mathcal{D}_j) + \epsilon_i(h^\star) + \epsilon_i^\star$$

$$\leq \epsilon_i(h^\star) + \epsilon_j(h^\star) + \epsilon_i^\star + \epsilon_j^\star + \frac{1}{2}d_{\mathcal{H}\Delta\mathcal{H}}(\mathcal{D}_i, \mathcal{D}_j)$$

The second line follows from Lemma 3 from (Ben-David et al., 2010), and the third from the triangle inequality. From this and proposition 1 we obtain the result. $\square$

### Corollaries for the 2-domain case

**Corollary 4.** *Given a domain $\mathcal{X}$, two distributions $\mathcal{D}_S$ and $\mathcal{D}_T$ over $\mathcal{X} \times \{0, 1\}$ and a hypothesis class $\mathcal{H}$ on $\mathcal{X}$, we have for $h \in \mathcal{H}$*

$$|\epsilon_S(h) - \epsilon_T(h)| \leq \epsilon_T^\star + \epsilon_S^\star + \Delta + d_{\mathcal{H}}(\mathcal{D}_S^X, \mathcal{D}_T^X) \tag{8}$$

*with $\Delta = max(E_{\mathcal{D}_T^X}|h_S^\star(\mathbf{x}) - h_T^\star(\mathbf{x})|, E_{\mathcal{D}_S^X}|h_S^\star(\mathbf{x}) - h_T^\star(\mathbf{x})|)$*

**Corollary 5.** *Given a domain $\mathcal{X}$, two distributions $\mathcal{D}_S$ and $\mathcal{D}_T$ over $\mathcal{X} \times \{0; 1\}$ and a hypothesis class $\mathcal{H}$ on $\mathcal{X}$, we have for $h \in \mathcal{H}$*

$$|\epsilon_S(h) - \epsilon_T(h)| \leq 2(\epsilon_T^\star + \epsilon_S^\star) + \beta + \frac{1}{2}d_{\mathcal{H}\Delta\mathcal{H}}(\mathcal{D}_S, \mathcal{D}_T) + d_{\mathcal{H}}(\mathcal{D}_S^X, \mathcal{D}_T^X) \tag{9}$$

*where*

$$\beta = \epsilon_S(h^\star) + \epsilon_T(h^\star) = \min_{h \in \mathcal{H}} \epsilon_S(h) + \epsilon_T(h)$$

## C CELL DATASET

### C.1 TEXAS DOMAIN

This dataset is extracted from that published in (Kang et al., 2016). It contains 455 biologically active images, in 11 classes, on four 384-well plates, in three channels: H2B-CFP, XRCC5-YFP and cytoplasmic-mCherry. Our analysis used 10 classes: 'Actin', 'Aurora', 'DNA', 'ER', 'HDAC', 'Hsp90', 'MT', 'PLK', 'Proteasome', 'mTOR'.

On top of the quality control from the original paper, a visual quality control was implemented to remove images with only apoptotic cells, and XRCC5-YFP channel images were smoothed using a median filter of size 2 using SciPy (Jones et al., 2001–).

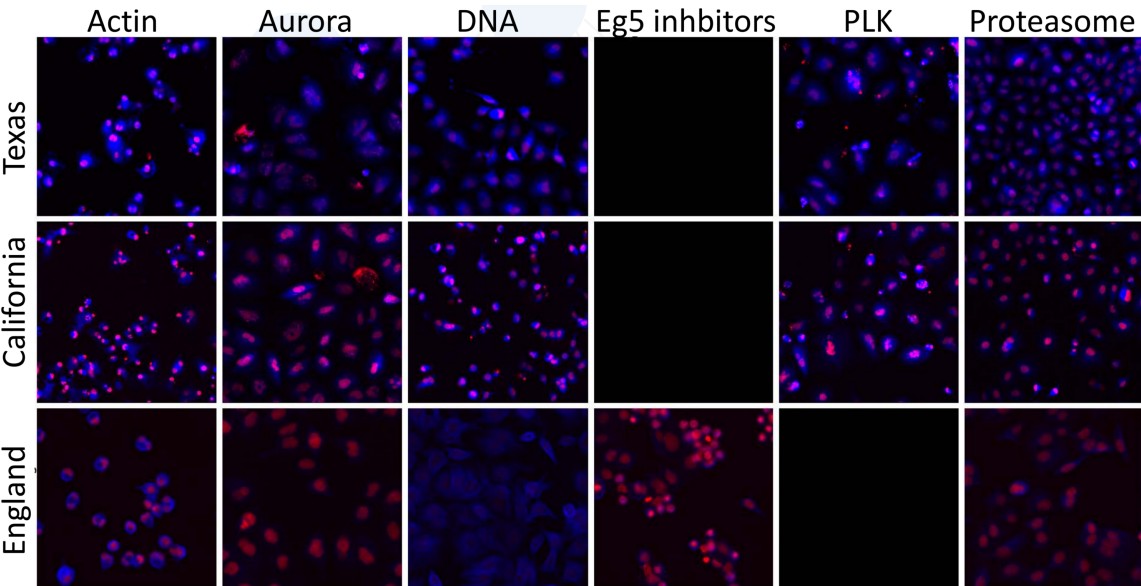

Figure 2: Examples from six classes in the Bio dataset (red: cell nuclei, blue: cell cytoplasm, magnification: 10X). Empty squares: the domain does not contain any known examples from this class. Best seen in color.

## C.2 CALIFORNIA DOMAIN

This dataset is designed to be similar to the Texas domain (Kang et al., 2016), generated using the same cell line, but in a different laboratory, by a different biologist, and using different equipment. It contains 1,077 biologically active images, in 10 classes, on ten 384-well plates, in three channels: H2B-CFP, XRCC5-YFP and cytoplasmic-mCherry. The classes are: 'Actin', 'Aurora', 'DNA', 'ER', 'HDAC', 'Hsp90', 'MT', 'PLK', 'Proteasome', 'mTOR'.

**Cell culture, drug screening and image acquisition**  Previously (Kang et al., 2016), retroviral transduction of a marker plasmid "pSeg" was used to stably express H2B-CFP and cytoplasmic-mCherry tags in A549 human lung adenocarcinoma cells. A CD-tagging approach (Sigal et al., 2006) was used to add an N-terminal YFP tag to endogenous XRCC5.

Cells were maintained in RPMI1640 media containing 10% FBS, 2 mM glutamine, 50 units/ml penicillin, and 50 $\mu$g/ml streptomycin (all from Life Technologies, Inc.), at $37^\circ C$, 5% $CO^2$ and 100% humidity. 24h prior to drug addition, cells were seeded onto 384-well plate at a density of 1200 cells/well. Following compound addition, cells were incubated at $37^\circ C$ for 48 hours. Images were then acquired using a GE InCell Analyzer 2000. One image was acquired per well using a 10x objective lens with 2x2 binning.

**Image processing**  Uneven illumination was corrected as described in (Stoeger et al., 2015). Background noise was removed using the ImageJ RollingBall plugin (Schneider et al., 2012). Images were segmented, object features extracted and biological activity determined as previously described (Kang et al., 2016). A visual quality control was implemented to remove images with obvious anomalies (e.g. presence of a hair or out-of-focus image) and images with only apoptotic cells. YFP-XRCC5 channel images were smoothed using a median filter of size 2.

## C.3 ENGLAND DOMAIN

This dataset was published by Caie et al. (2010) and retrieved from (Ljosa et al., 2012). It contains 879 biologically active images of MCF7 breast adenocarcinoma cells, in 15 classes on 55 96-well plates, in 3 channels: Alexa Fluor 488 (Tubulin), Alexa Fluor 568 (Actin) and DAPI (nuclei). Classes with fewer than 15 images and absent from the other datasets ("Calcium regulation", "Cholesterol",

"Epithelial", "MEK", "mTOR") were not used, which leaves 10 classes: 'Actin', 'Aurora', 'DNA', 'ER', 'Eg5 inhibitor', 'HDAC', 'Kinase', 'MT', 'Proteasome', 'Protein synthesis'.

**Image processing**  As the images were acquired using a 20X objective, they were stitched using ImageJ plugin (Preibisch et al., 2009) and down-scaled 2 times. Cells thus appear the same size as in the other domains. Images were segmented, object features extracted and biological activity obtained as previously described (Kang et al., 2016). A visual quality control was implemented to remove images with obvious anomalies and images with only apoptotic cells. Images with too few cells were also removed: an Otsu filter (Otsu, 1979) was used to estimate the percentage of pixels containing nuclei in each image, and images with less than 1% nuclear pixels were removed. Tubulin channel images were smoothed using a median filter of size 2.

## C.4   COMMON IMAGE PRE-PROCESSING

Images which were not significantly distinct from negative controls were identified as previously (Kang et al., 2016) and excluded from our analysis. Previous work on the England dataset further focused on images which "clearly [have] one of 12 different primary mechanims of action" (Ljosa et al., 2012). We chose not to do so, since it results in a simpler problem (90% accuracy easy to reach) with much less room for improvement.

Images from all domains were down-scaled 4 times and flattened to form RGB images. Images were normalized by subtracting the intensity values from negative controls (DMSO) of the same plate in each channel. England, Texas and California share images for cell nucleus and cytoplasm, but their third channel differs: Texas and California shows the protein XRCC5, whereas England shows the Actin protein. Therefore, the experiments which combine Texas and England, and California and England used only the first two channels, feeding an empty third channel into the network. Similarly, profiles contain 443 features which are related to the first two channels, and 202 features which are related to the third channel. Only the former were used in experiments which involve the England dataset.

## C.5   SEMI-SUPERVISED MDL EXPERIMENTS

| Shift | Dom. 2, labeled classes | Domain 2, unlabeled classes |
|-------|-------------------------|------------------------------|
| E-C   | HDAC, Proteasome, Actin, Aurora | DNA, MT, ER |
| C-T   | DNA, HDAC, MT, ER, Aurora, mTOR, PLK | Actin, Proteasome, Hsp90 |
| T-E   | DNA, MT, Proteasome, Actin, ER | Aurora, HDAC, Actin |
| C-T-E | DNA, MT, Proteasome, Actin, ER | Aurora, HDAC, Actin |

Table 3: Class content for the CELL experiments in table 2. In all cases, the first domain contains the same classes as domain 2, though with labeled examples from all classes. These classes were picked as those with best classification accuracy in an unsupervised setting; results are similar when picking the classes with worst classification accuracy. 10 labeled images per class were used for training.

## D  EXPERIMENTAL SETTINGS

### D.1  ARCHITECTURE

As in (Ganin et al., 2016; Tzeng et al., 2014), a bottleneck fully connected layer is added after the last dense layer of VGG-16. Learning rates on weights (resp. biases) from "from scratch" layers is ten (resp, twenty) times that on parameters of fine-tuned layers. Instance normalization is used on DIGITS, whereas global normalization is used on OFFICE and CELL.

| OFFICE and CELL | DIGITS |
|---|---|
| Feature extractor | |
| VGG-16, layers $Conv_1$ to $FC_7$ | 5x5 conv. 32; ReLU; 2x2 max pool, 2x2 stride |
| Fully connected 256 | 5x5 conv. 48; ReLU; 2x2 max pool, 2x2 stride |
| Classifier | |
| Output of feature extractor | Output of feature extractor |
| | Fully connected 100; ReLU |
| | Fully connected 100; ReLU |
| Fully connected $L$; Softmax | Fully connected $L$; Softmax |
| Domain discriminator | |
| Output of feature extractor | Output of feature extractor |
| Gradient reversal layer | Gradient reversal layer |
| Fully connected 1,024; ReLU; Dropout (0.5) | Fully connected 100; ReLU |
| Fully connected 1,024; ReLU; Dropout (0.5) | |
| Fully connected $i$; Activation | Fully connected $i$; Activation |

Table 4: Architectures. In the case when considering only two domains, $i = 1$ and the last activation of domain discriminators is a sigmoid. When considering three domains, $i = 3$ and the activation is a softmax. Knowledge discriminator architecture is identical to that of domain discriminators without the gradient reversal layer.

### D.2  HYPER-PARAMETER SEARCH

| Parameter | DIGITS and Signs | CELL |
|---|---|---|
| Learning rate (lr) | $10^{-3}, 10^{-4}$ | $10^{-4}$ (+ $10^{-5}$ for 3-dom.) |
| Individual lr | NA | True, False |
| Lr schedule | Exponentially decreasing, constant | |
| $\lambda$ | 0.1, 0.8 | |
| $\lambda$ schedule | Exponentially increasing, constant | |
| $\zeta$ | 0.1, 0.8 | |

Table 5: Range of hyper-parameters which were evaluated in cross-validation experiments. Exponentially decreasing schedule, exponentially increasing schedule, indiv. lr (learning rates from layers which were trained from scratch are multiplied by 10), as in (Ganin et al., 2016).

# E    ADDITIONAL RESULTS

## E.1    3-DOMAIN RESULTS ON OFFICE

Table 6: Classification results on target test set in the semi-supervised DA setting (average and stdev on 5 seeds or folds)

| Sources | D, W | A, W | A, D |
|---|---|---|---|
| Target | Amazon | DSLR | Webcam |
| Baseline | 41.7 (1.0) | 90.9 (1.3) | 89.4 (1.5) |
| DANN | 57.5 (1.6) | 92.3 (1.8) | 91.2 (0.7) |
| FT  MADA | 37.5 (6.8) | Not conv. | 88.3 (0.7) |
| MuLann | 54.5 (3.8) | 92.1 (2.6) | 92.0 (1.0) |

|         DANN         |         MADA         |         MuLANN         |
|:--------------------:|:--------------------:|:----------------------:|

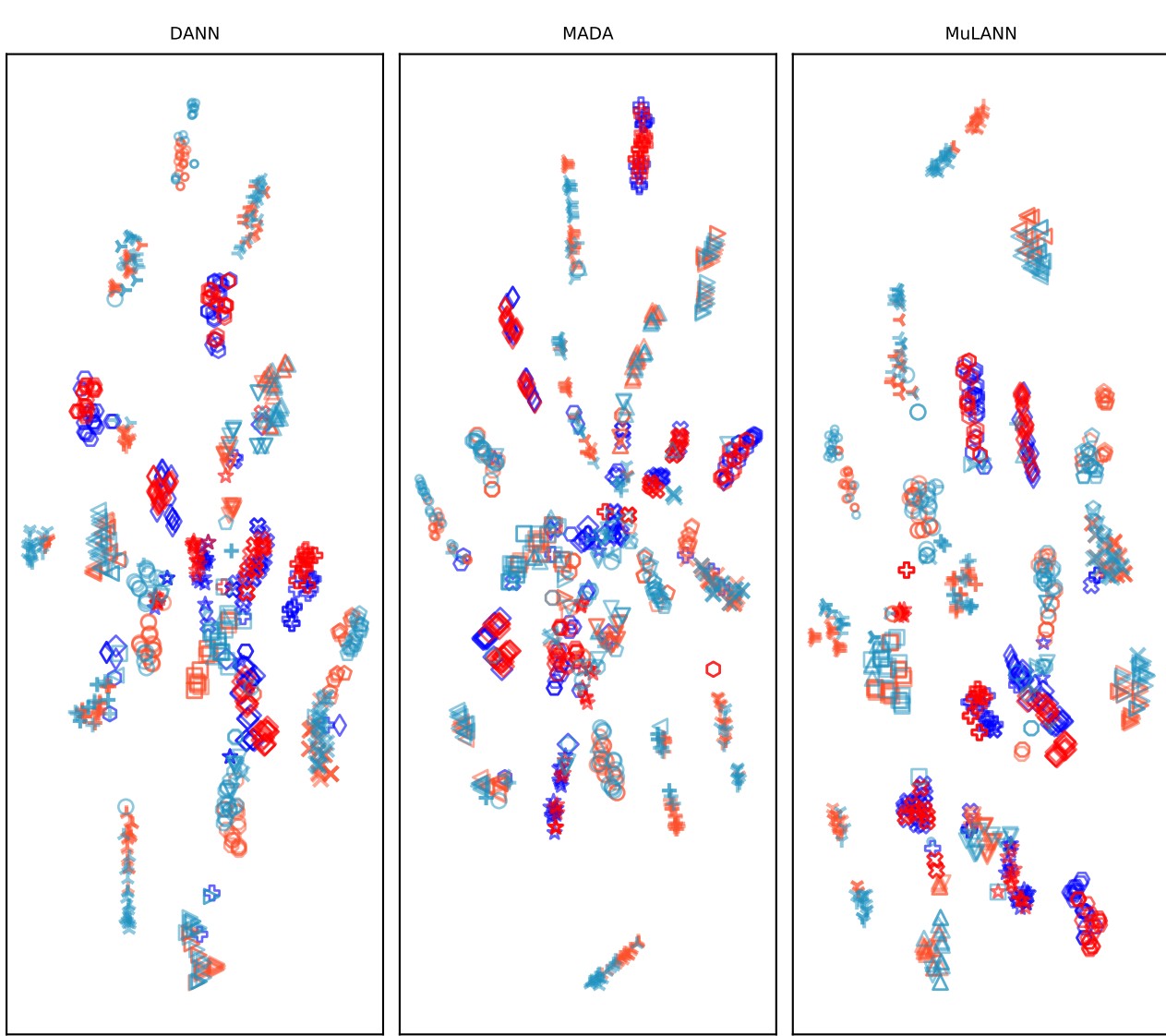

Figure 3: Visualization of class features on Webcam (red) > Amazon (blue). Dimmer colors indicate classes for which labeled examples are available in both domains.

### E.2   tSNE VISUALIZATION

We use tSNE (van der Maaten & Hinton, 2008) to visualize the common feature space in the example of Webcam $\rightarrow$ Amazon. Fig. 3 shows that classes are overall better separated with MULANN. In particular, when using MULANN, unlabeled examples (blue) are both more grouped and closer to labeled points from the other domain.

### E.3 SEMI-SUPERVISED MDL ON THE BIO DATASET

Table 5: CELL average test classification results on all domain (average and stdev on 5 folds). P stands for "profiles", "lab." for labeled and "unlab." for unlabeled. Baselines are obtained by training MULANN with $\lambda = 0$ (NN) and LDA+k-NN (P) on both domains. Results were obtained in the non-fully transductive setting, without hyper-parameter optimization.

| Shift | Image set | # classes | Baseline NN | DANN | MADA | MULANN | Baseline P | P+Coral |
|---|---|---|---|---|---|---|---|---|
| | E | 7 | 74.1 (5.4) | 71.6 (5.8) | 63.6 (6.1) | 72.7 (4.0) | 78.1 (8.0) | 66.4 (2.4) |
| E-C | C lab. | 4 | 98.3 (0.6) | 96.1 (1.5) | 92.3 (5.2) | 89.1 (6.4) | 98.2 (2.4) | 94.1 (2.3) |
| | C unlab. | 3 | 0.4 (0.7) | 34.8 (20.7) | 14.5 (7.4) | 25.7 (12.3) | 21.5 (8.4) | 36.8 (3.7) |
| | C | 10 | 91.4 (1.8) | 87.0 (2.2) | 87.9 (3.9) | 89.3 (1.8) | 96.1 (1.1) | 93.3 (1.8) |
| C-T | T lab. | 7 | 93.7 (1.3) | 91.0 (4.4) | 86.7 (7.5) | 89.2 (1.2) | 96.2 (2.4) | 92.8 (3.2) |
| | T unlab. | 3 | 24.4 (10.0) | 61.4 (7.7) | 56.2 (14.0) | 77.7 (4.0) | 59.6 (11.3) | 87.6 (8.2) |
| | T | 7 | 95.2 (2.2) | 90.3 (5.4) | 93.7 (3.0) | 88.2 (6.4) | 94.2 (6.3) | 92.6 (4.0) |
| T-E | E lab. | 4 | 75.2 (9.7) | 61.9 (8.5) | 71.0 (12.7) | 72.8 (14.2) | 81.1 (8.8) | 61.2 (4.0) |
| | E unlab. | 3 | 5.7 (6.6) | 31.4 (17.5) | 26.0 (19.4) | 51.3 (13.5) | 16.1 (9.5) | 25.7 (12.6) |
| | C | 7 | 94.7 (2.0) | 91.7 (1.4) | 82.7 (3.8) | 93.9 (1.7) | 94.1 (2.0) | 89.4 (2.2) |
| C-T-E | T | 7 | 94.8 (2.1) | 93.7 (4.7) | 86.5 (4.2) | 94.9 (2.1) | 97.8 (0.5) | 89.6 (8.0) |
| | E lab. | 4 | 74.1 (9.8) | 67.7 (12.8) | 48.2 (11.7) | 66.6 (9.0) | 74.7 (10.5) | 55.6 (7.5) |
| | E unlab. | 3 | 10.7 (9.7) | 48.6 (21.9) | 22.6 (11.3) | 69.3 (21.1) | 36.3 (6.6) | 52.5 (22.5) |

### E.4 IMPACT OF $p - p^\star$ ON A DOMAIN WITHOUT UNLABELED DATAPOINTS

### E.5 ASYMMETRY RESULTS ON CELL

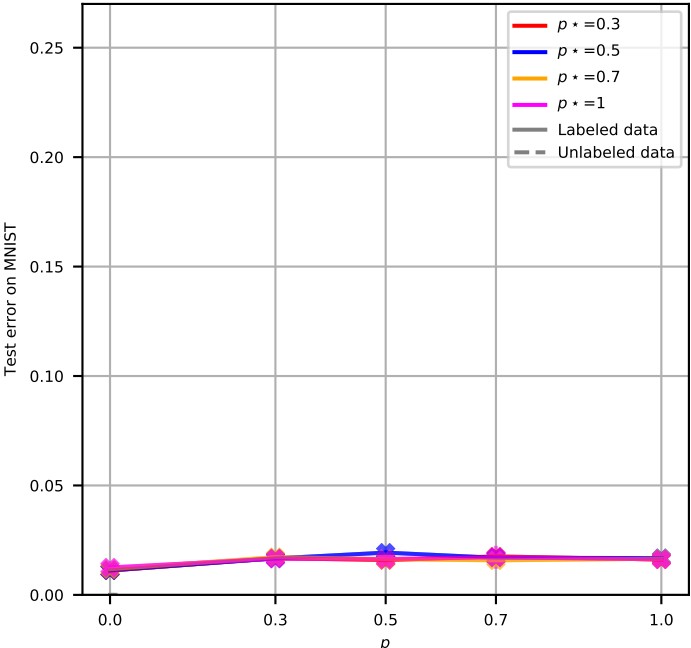

Figure 4: Impact of parameter $p$ in comparison with $p^\star$ on MNIST $\leftrightarrow$ MNIST-M. $p = 0$ corresponds to DANN (see text for details): no data flowed through the KUD module. We can see that different values of $(p, p^\star)$ do not influence the accuracy on a domain which did not have any unlabaled datapoints from extra classes (MNIST in this case).

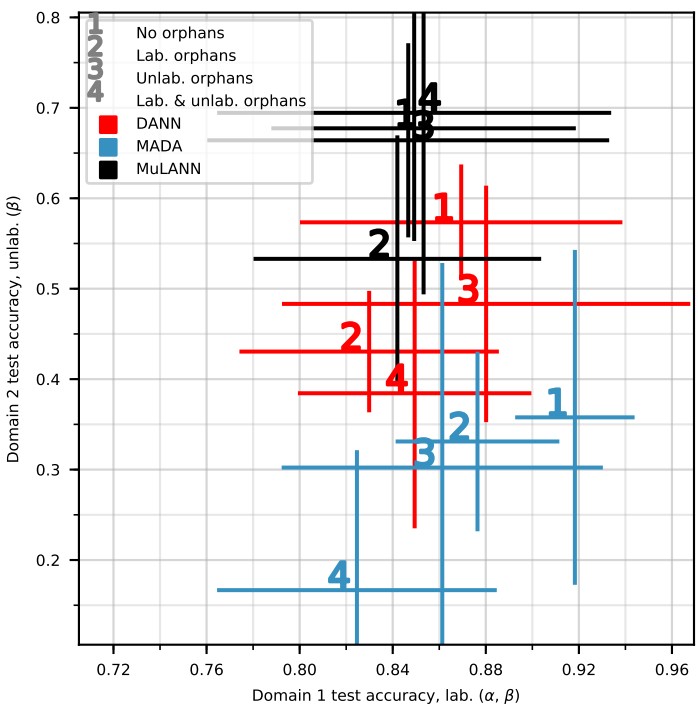

Figure 5: Impact of asymmetry in class content between domains on CELL (T↔E) for DANN, MADA and MULANN.

