# OpenReview forum: "Multi-Domain Adversarial Learning"
_ICLR.cc/2019/Conference_

### Official Review · AnonReviewer3 · 2018-11-02
**Interesting known/unknown label multidomain learning setting, but weak evaluation**

**Rating:** 6
**Confidence:** 5

**Review:**

Summary:

The manuscript proposes a multi-domain adversarial learning (MDL) method called MULANN, to leverage multiple datasets with overlapping but distinct class sets, in a semi-supervised setting. The authors define a new discrimination task to discriminate, within each domain, labeled samples from unlabeled ones that most likely belong to extra classes (classes with no labeled or unlabeled samples in the domain). They also introduce a bound on the average- and worst-domain risk in MDL, obtained using the H-divergence.

Strengths:

- The idea of using discriminators for separating the labeled samples from unlabeled ones that most likely belong to extra classes is interesting.

- A new generalization bound for MDL is introduced.

- The paper was clear, well written, well-motivated and nicely structured.

- The authors perform numerous empirical experiments on several types of problems on various datasets (Digit, OFFICE,CELL) successfully showing how the MULANN can reduce the nasty effects of the adversarial domain discriminator and repulse (a fraction of) unlabeled examples from labeled ones in each domain.

Weaknesses:

- all the experiments except the last row of Table 2 concern adaptation between two domains. Given the paper title, the reviewer would have expected more experiments in a multiple domain context. More precisely, for the digit datasets, the reviewer was interested to see how the proposed MDL performs on jointly adapting SVHN, MNIST, MNIST-M, and USPS or jointly adapting DSLR, Amazon, and Webcam for OFFICE dataset. Moreover, comparison with some of the DA baselines (ADDA[1], DSN[2]) is missing.

- The authors propose to rank the unlabeled samples of each domain according to the entropy of their classification of the current classifier. Obviously there must be some false ranking (specially at the initial stages of updating the classifier) for the unlabeled samples (e.g. the classifier may output high entropy for the unlabeled samples of the classes with labeled samples) and they may harm the performance of adaptation.  It is not clear how MULANN can work in this situation and how its performance vary with the noisy signals conveyed in those false pseudolabeled samples.

- Although the paper introduces the generalization bound for MDL, it does not give new formulation or algorithm to handle MDL (MULANN handles only the class asymmetry when domains involve distinct sets of classes and it has nothing to do with MDL). hence, there is no connection between the theoretical results on MDL generalization bound and the proposed method MULANN.

- Since each domain may have different number of classes, it is not clear how the number of classes (L) is set in the classification module (maximum number of classes in all domain?).

The reviewer is also interested to see how the the generalization bound introduced in this paper is related to the recent theoretical works [3],[4] on MDL.

[1] Tzeng, Eric, et al. "Adversarial discriminative domain adaptation." Computer Vision and Pattern Recognition (CVPR). Vol. 1. No. 2. 2017.

[2] Bousmalis, Konstantinos, et al. "Domain separation networks." Advances in Neural Information Processing Systems. 2016.

[3] Zhao, Han, et al. "Multiple Source Domain Adaptation with Adversarial Learning." Advances in Neural Information Processing Systems. 2018.

[4] Hoffman, Judy, Mehryar Mohri, and Ningshan Zhang. "Algorithms and Theory for Multiple-Source Adaptation."  Advances in Neural Information Processing Systems. 2018.

---

> ### Author Response · Authors · 2018-11-17
> **Connection between the theoretical results and proposed approach, addition of 3-domain experiments on Office and commenting on recent multiple source DA papers**
>
> We thank the reviewer for their insightful comments.
>
> Q1 "all the experiments except the last row of Table 2 concern adaptation between two domains. Given the paper title, the reviewer would have expected more experiments in a multiple domain context."
>
> A1 A main difference between domain adaptation and MDL is the fact that the former aims to minimize the target error, while the latter aims to minimize the average error. In this sense, our goal (and the validation experiments on Cell) are focused on MDL.
>
> Q2 "Although the paper introduces the generalization bound for MDL, it does not give new formulation or algorithm to handle MDL" [...] "There is no connection between the theoretical results on MDL generalization bound and the proposed method MULANN."
>
> A2 This issue is related to the above: the new generalization bound extends that of Ben David et al. in the sense that it considers all pairs of domains involved, thus bounding the *average* risk; and this bound is the one underlying the proposed algorithm and its MDL experiments. We have clarified this in the manuscript.
>
> Q3 "the reviewer was interested to see how the proposed MDL performs on jointly adapting SVHN, MNIST, MNIST-M, and USPS or jointly adapting DSLR, Amazon, and Webcam for OFFICE dataset. "
>
> A3: We added 3 domain experiments for Office, which are now displayed in Appendix E.1 table 6. As discussed in [3], we also find that the addition of a second source is not necessarily beneficial to target accuracy.
>
> Q4: Comparison with some of the DA baselines (ADDA[1], DSN[2]) is missing."
> A4: ADDA, an unsupervised DA method, proceeds by training sequentially a classifier on Source, then learning the Target feature space by making it indistinguishable from the Source one. However this is not applicable to the semi-supervised setting: either target labels would not be used in the first training step, or they would be used but without any domain loss to account for the fact that two domains are being used at the same time. Thus, the classifier would actually learn two sub-classifiers: one for each domain, which would turn counter-productive in the second step where this strong distinction between source and target would have to be un-learned.
> We are re-programming DSN and experimental results will be added. We thank the reviewer for the suggestion.
>
> Q5: "The reviewer is also interested to see how the the generalization bound introduced in this paper is related to the recent theoretical works [3],[4] on MDL."
>
> A5 Zhao et al. [3] consider the multiple source context; they define a weighted scheme where the weight of a source depends on its H-divergence with the target, plus its own classification error. The feature extractor is trained either from the best source only (in the sense of this weight), or from a weighted sum of the sources. When interested in multi-domain learning (thus aiming to minimize the average risk), it seems that there are two possibilities: a single feature extractor; or a feature extractor per domain. In the former case, the feature extractor might be overly conservative; in both cases, scalability w.r.t. the number of domains might be an issue.
>
> Hoffman et al. [4] also consider the multiple source context, assuming that the target is a unknown mixture of the sources (or not too far thereof in terms of Renyi divergence). Their experiments follow this assumption (using as target a mixture of sources Amazon, Webcam and DSLR). In our case this assumption does not hold, e.g. the joint distribution of England(x,y) is *not* a mixture of Texas(x,y) and California(x,y) (as can be seen by eye, and confirmed by experiments). The adversarial change of representation only enforces the merge of the marginals.
>
> Q6 "The authors propose to rank the unlabeled samples of each domain according to the entropy of their classification of the current classifier. Obviously there must be some false ranking (specially at the initial stages of updating the classifier) for the unlabeled samples"
>
> A6 As the reviewer suggests, there are indeed misclassifications of samples using their entropy ranking in early training stages. We mention this section 4.3. This misclassification is the reason why it is better that hyper-parameter p slightly underestimates p* than is equal to it, as can be seen in Fig. 1, right (except when p*=1 as one could expect).
>
> Q7 "Since each domain may have different number of classes, it is not clear how the number of classes (L) is set in the classification module (maximum number of classes in all domain?)."
>
> A7 L is the cardinal of the union of classes with labeled examples in at least one domain.

---

### Official Review · AnonReviewer1 · 2018-11-04
**This paper has good presentation and sufficient contribution to the field**

**Rating:** 8
**Confidence:** 5

**Review:**

In this paper, the authors proposed a multi-domain adversarial learning approach, MULANN, to improve the classification accuracy on three datasets-DIGITS, OFFICE and CELL-in the semi-supervised DA setting. It’s contributions include: i) using the H-divergence to bound both the risk across all domains and the worst-domain risk (imbalance on a specific domain); ii) a new loss to accommodate semi-supervised multi-domain learning and domain adaptation; iii) the experimental validation of the approach, improving on the state-of-the-art on two standard image benchmarks, and a novel bioimage dataset, CELL.
In addition, this paper has a clear logic to explain and prove the problem to be solved, and has ample experimental evidence. Above on, this paper did a meaningful work. But there are some errors of expression, so it should be checked.

---

> ### Author Response · Authors · 2018-11-17
> **Thank you**
>
> We thank the reviewer for their encouraging words and will correct the errors of expression.

---

### Official Review · AnonReviewer2 · 2018-11-05
**Good idea, but the results are not particularly convincing**

**Rating:** 5
**Confidence:** 4

**Review:**

PROS:
* Original idea of using separate "discriminator" paths for unknown classes
* Thorough theoretical explanation
* A variety of experiments
* Very well-written, and clear paper

CONS:
* The biggest problem for me was the unconvincing results. MNIST-to-MNIST-M has better baselines  (PixelDA performed better on this task for example), Office is not suitable for domain adaptation experiments anymore unless one wants to be in a few-datasample regime or work with data with noisy labels(the dataset is plagued with label pollution, and there are too few examples per class per domain for NN-based domain adaptation); the results on CELL were not convincing, I don't know the dataset but it seems that baseline NN does better than DA most of the times.
* Comparison with other methods did not take into account a variety of hyperparameters. Although I do understand the problem of evaluation in unsupervised DA, this should have at least been done in the semi-supervised case, and some analysis/discussion should be included for the unsupervised one. What if the proposed method performs that much better than baselines but they hyperparameters are not set correctly?

---

> ### Author Response · Authors · 2018-11-17
> **Updated results with hyper-parameter optimization, overall better than existing methods**
>
> We thank the reviewer for their encouraging words.
>
> "There are better baselines for MNIST to MNIST-M": yes, the presented method might not outperform all other methods on all other datasets. Still, we hope to convince the reviewer that the results on the other datasets are worth being considered.
>
> "Office is not suitable unless one wants to be in a few-datasample regime or work with data with noisy labels": we would like to point that this regime is quite realistic in Bioimage informatics (noisy, with few samples per class).
>
> "The results on Cell are not convincing": as our goal is multi-domain learning on this dataset, the relevant performance indicator is the average risk over all domains. Table 2 details what happens in various categories of cases (on classes with/without labelled samples). Despite the (well-known) degradation of the results on labeled classes when one also considers unlabelled classes, the bottom line is that -- regarding the average risk -- our method outperforms the baseline. Importantly, MuLANN results on Cell are significantly better than the baseline on all rows which involve unlabeled classes (rows  3, 6, 9, 13), while remaining not significantly different to the baseline on 6/9 of the other rows.
>
> "Comparison with other methods did not take into account a variety of hyperparameters". The reviewer is right. Complementary experiments have thus been performed, and tables 1, 2 updated. We investigated the impact of varying the learning rate, the weight lambda on the discriminator loss, the weight dzeta of the known-unknown discrimination loss, the learning rate schedule, lambda schedule as well as using different learning rates for pre-trained layers versus from scratch layers (see Table 5 for more detailed information). These results show a moderate sensitivity of MuLANN, MADA and DANN wrt hyper-parameters and confirm that MuLANN outperforms both MADA and DANN (detailed results available here https://drive.google.com/file/d/1NjtMKF53qmnx4_Jyvh-ofxb0WjzcDvow/view?usp=sharing).

---

### Meta-Review · Area_Chair1 · 2018-12-12
**Theoretical contribution limiting worst case performance in the multi-domain setting for adversarial based adaptation methods**

**Confidence:** 5
**Recommendation:** Accept (Poster)

**Metareview:**

This paper extends the single source H-divergence theory for domain adaptation to the case of multiple domains. Thus, drawing on the known connection between H-divergence and learning the domain classifier for adversarial adaptation, the authors propose a multi-domain adversarial learning algorithm. The approach builds upon the gradient reversal version of adversarial adaptation proposed by Ganin et al 2016.

Overall, multi-domain learning and limiting the worst case performance on any single domain is an interesting problem which has been relatively underexplored. Though this work does not have the highest performance on all datasets across competing methods, as noted by reviewers, it proposes a useful theoretical result which future research may build on. I would encourage the reviewers to compare against and discuss the missing prior work cited by Rev 3.